# LEARNING POWERFUL POLICIES AND BETTER DYNAMICS MODELS BY ENCOURAGING CONSISTENCY

## ABSTRACT

Model-based reinforcement learning approaches have the promise of being sample efficient. Much of the progress in learning dynamics models in RL has been made by learning models via supervised learning. There is enough evidence that humans build a model of the environment, not only by observing the environment but also by interacting with the environment. Interaction with the environment allows humans to carry out *experiments*: taking actions that help uncover true causal relationships which can be used for building better dynamics models. Analogously, we would expect such interaction to be helpful for a learning agent while learning to model the environment dynamics. In this paper, we build upon this intuition, by using an auxiliary cost function to ensure consistency between what the agent observes (by acting in the real world) and what it imagines (by acting in the "learned" world). Our empirical analysis shows that the proposed approach helps to train powerful policies as well as better dynamics models.

## 1 INTRODUCTION

Reinforcement Learning consists of two fundamental problems: *learning* and *planning*. *Learning* comprises of improving the agent's current policy by interacting with the environment while *planning* involves improving the policy without interacting with the environment. These problems evolve into the dichotomy of *model-free* methods (which primarily rely on *learning*) and *model-based* methods (which primarily rely on *planning*). Recently, *model-free* methods have shown many successes, such as learning to play Atari games with pixel observations (Mnih et al., 2015b; Mnih et al., 2016) and learning complex motion skills from high dimensional inputs (Schulman et al., 2015a;b). But their high sample complexity is a still a major criticism of the *model-free* approaches.

In contrast, model-based reinforcement learning methods have been introduced in the literature where the goal is to improve the sample efficiency by learning a dynamics model of the environment. But model-based RL has several caveats. If the policy takes the learner to an unexplored state in the environment, the learner's model could make errors in estimating the environment dynamics, leading to sub-optimal behaviour. This problem is referred to as the model-bias problem (Deisenroth & Rasmussen, 2011).

In order to make prediction about the future, dynamics models are unrolled step by step which leads to the issue of "compounding errors" (Talvitie, 2014; Bengio et al., 2015; Lamb et al., 2016): an error in modeling the environment at time $t$ affects the predicted observations at all subsequent time-steps. This problem is much more challenging for the environments where the agent observes high-dimensional image inputs and not compact state representations. On the other hand, model-free algorithms are not limited by the accuracy of the model, and therefore can achieve better final performance by trial and error, though at the expense of much higher sample complexity. In the model-based approaches, the dynamics model is usually trained with supervised learning techniques mostly just by *observing the data*. On the other hand, there's enough evidence that humans learn the environment dynamics not just by observing the environment but also by interacting with the environment (Cook et al., 2011; Daniels & Nemenman, 2015).

This leads to an interesting possibility. The agent could consider two possible pathways: (i) Taking actions in the real world to generate new observations and (ii) Imagining to take actions and predicting the new observations. Consider the humanoid robot from the MuJoCo environment (Mordatch et al., 2015). In the first case, the humanoid agent takes an action in the real environment, observes

the change in its position (and location), takes another step and so on. In the second case, the agent imagines taking a step, predicts what the observation would look like, imagines taking another step and so on. The first case is a *close-loop* setup, where the humanoid observes the state of the world, takes an action, gets the true observation from the environment, which is used to choose the next action, and so on. The second case is a *open-loop* setup, where the agent predicts subsequent states for multiple time steps into the future without interacting with the environment. The two cases have been summarized in figure 1.

As such, the two pathways may not to be "consistent" given the challenges in learning a multi-step dynamics model. By "consistent", we mean the behaviour of state transitions along the two paths should be indistinguishable. Had they been consistent, the learner's model would be grounded in reality ie the predictions from the open loop would be similar to the predictions from the closed loop over a long time horizon. In this work, we propose to ensure consistency by using an auxiliary loss which explicitly seeks to match the generative behaviour (from open loop) and the observed behaviour (from closed loop) as closely as possible. More generally, we show that the proposed approach helps to simultaneously train more powerful policies as well as better dynamics models, by using a training objective that is not solely focused on predicting the next observation. Our evaluation protocol consists of learning both observation-space models and state-space models. We consider various continuous control tasks from the OpenAI Gym suite (Brockman et al., 2016), and RLLab (Duan et al., 2016) and show that using the proposed auxiliary loss consistently helps in achieving better performance across tasks. We evaluate the proposed approach on the pixel-based cheetah domain from the OpenAI Gym suite (Brockman et al., 2016). This domain is bit difficult for the "baseline" state space models as one can only infer positions and not velocities from the images and hence this makes the task partially observable. We compare the proposed model to the state of the art state space models (Buesing et al., 2018), and show that the proposed method consistently achieves better results.

## 2 PRELIMARIES

A finite time Markov decision process $\mathcal{M}$ is generally defined by the tuple $(\mathcal{S}, \mathcal{A}, f, R, \gamma)$. Here, $\mathcal{A}$ the action space, $\mathcal{S}$ is the set of states, $f(s_{t+1}|s_t, a_t)$ the transition distribution, $r : \mathcal{S} \times \mathcal{A} \to \mathbb{R}$ is the reward function and $\gamma$ the discount factor. We define the return as the discounted sum of rewards $r(s_t, a_t)$ along a trajectory $\tau := (s_0, a_0, ..., s_{T-1}, a_{T-1}, s_T)$, here $T$ refers to the effective horizon of the process. The goal of reinforcement learning is to find a policy $\pi_\phi$ that maximizes the expected return. Here $\phi$ denotes the parameters of the policy $\pi$.

Model-based RL methods learn the dynamics model from the observed transitions. This is usually done with a function approximator usually parameterized as a neural network $\hat{f}_\theta(s_{t+1}|s_t, a_t)$. In such case, the parameters $\theta$ of the dynamics model are optimized to maximize the log-likelihood of the state transition distribution.

## 3 ENVIRONMENT MODEL

Consider a learning agent which is training to optimize a reward signal $r$ in a given environment. At a given timestep $t$, the agent is in some state $s_t \in S$. It takes an action $a_t \in A$ according to its policy $a_t \sim \pi_t(a_t|s_t)$, receives a reward $r_t$ (from the environment) and transitions to a new state $s_{t+1}$. The agent is trying to maximize its expected reward and has two pathways for improving its behaviour:

1. ***Close-loop* path:** Here, the learner interacts with the environment at every step. The agent starts in state $s_0$ and is in state $s_t$ at time $t$. It chooses an action $a_t$ to perform (using its policy $\pi_t$), performs the chosen action, and receives a reward $r_t$. It then observes the environment to obtain the new state $s_{t+1}$, uses this state to decide which action $a_{t+2}$ to perform next and so on.

2. ***Open-loop* path:** Here, the learner predicts the future observations (or future belief state in case of state space models). The agent starts in state $s_0$ and in state $s_t$ at time $t$. Note that the agent "imagines" itself to be in state $s_t^I$ and can not access the true state of the environment. It chooses an action $a_t$ to perform (using its policy $\pi_t$), performs the action in the "learners" world and imagines to transition to the new state $s_{t+1}^I$. Thus the current

"imagined" state is used to predict the next "imagined" state. During these "imagined" roll-outs, the agent does not interact with the environment but interacts with its "imagined" version of the environment which we call its dynamics model or the learner's "world".

As an alternative, the agent could use both the pathways simultaneously. The agent could, in parallel, (i) Build a model of the environment and (ii) Engage in interaction with the environment as shown in Figure 1. We propose to make the two pathways consistent with each other so as to ensure that the predictions from the learner's dynamics model are grounded in the observations from the environment. We show that such a "consistency constraint" helps the agent to learn a powerful policy and a better dynamics model of the environment.

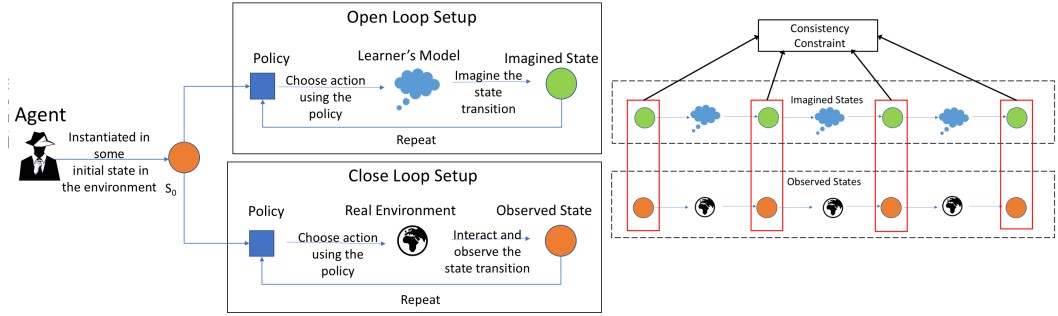

Figure 1: The agent, in parallel, (i) Builds a model of the world and (ii) Engages in an interaction with the world. The agent can now learn the model dynamics while interacting with the environment. We show that making these two pathways consistent helps in simultaneously learning a better policy and a more powerful generative model.

## 3.1 CONSISTENCY CONSTRAINT

We want the "imagined" behaviour (from the open loop) to be consistent with the observed behaviour (from the close loop). Thus making sure that the predictions from the learner's dynamics model (or the "world") are similar to the actual observations from the environment. The learner's dynamics model could either be in observation space (pixel space) or it could be in state space. State space models are generally more efficient as they model dynamics at some higher level of abstraction. In the case of state space models, the learner predicts transitions in the state space by first encoding the actual observation from the environment into the state space of the learner and then imposing the consistency constraint in the (learned) state space.

In order to apply the consistency constraint in the open loop setup, we need to obtain the corresponding state transitions in the environment. At a given timestep $t$, the learner is in some environment state $s_t$ while it imagines to be in state $s_t^I$. It takes an action $a_t$ according to its policy $a_t \sim \pi_t(a_t|s_t)$. Now the learner can make transition in two ways. It could execute the action in the environment and transition to state $s_{t+1}$ (as governed by the dynamics of the environment). Alternatively, it could execute the action in the "learned" dynamics environment $\hat{f}_\theta$ and imagine to transition to the state $s_{t+1}^I = \hat{f}_\theta(s_t^I, a_t)$. Note that the state $s_t$ is not used by the learner's dynamics model when making state transitions during the open-loop setup.

Many possibilities exist for imposing the "consistency constraint". In a simple setup, we could enforce the per step output from the open and the closed loops to be similar. The downside of this approach is that it encourages the dynamics model to mimic each and every detail of the environment irrespective of its utility in predicting the state transitions (Lamb et al., 2016). Hence, we encode the state transitions (during both open-loop and closed-loop) into fixed-size real vectors using recurrent networks and enforce the output of the recurrent networks to be similar in the two cases. Encoding the sequence can be seen as abstracting out the per-step state transitions into how the dynamics of the environment evolve over time. This way, we do not focus on mimicking each state but the high level dynamics of the state transitions. This consistency loss encourages the dynamics model to only focus on information that makes the multi-step predictions (from the open-loop) indistinguishable from

the actual future observations from the environment. This is shown in figure 1. Once we have the predicted state transitions and the real state transitions, we could impose the "consistency constraint" in several ways. In order to keep our setup simple, we minimize the prediction error ($L^2$ error) between the encoding of predicted future observations as coming from the learner's dynamics model (during open-loop) and the encoding of the future observations as coming from the environment (during closed loop).

Let us assume that the agent started in state $s_0$ and that $a_{0:T-1}$ denote the sequence of actions that the agent takes in the environment from time $t = 0$ to $T - 1$. Similarly, $s_{1:T}$ denotes the sequence of states that the agent transitions through. Alternatively, the agent could have "imagined" a trajectory of state transitions by performing the actions $a_{0:T-1}$ in the learner's dynamics model. This would result in the sequence of states $s_{1:T}^I$. The consistency loss is computed as follows:

$$enc(s_{1:T})) = RNN([s_1, s_2, ..., s_T])$$

$$enc(s_{1:T}^I)) = RNN([s_1^I, s_2^I, ..., s_T^I])$$

$$l_{cc}(\theta, \phi) = \|enc(s_{1:T})) - enc(s_{1:T}^I))\| \tag{1}$$

where $\|\|$ denotes the L2 norm and $RNN$ denotes the GRU model used to encode the sequence of state transitions into a fixed length vector.

The agent which is trained with the consistency constraint is referred to as the *consistent dynamics* agent. The overall loss for such a learning agent can be written as follows:

$$l_{total}(\theta, \phi) = l_{rl}(\theta, \phi) + \alpha l_{cc}(\theta, \phi) \tag{2}$$

where $\theta$ refers to the parameters of the agent's transition model $\hat{f}$ and $\phi$ refers to the parameters of the agent's policy $\pi$. The first component of the loss function, $l_{rl}(\theta, \phi)$, corresponds to the RL objective i.e maximizing expected return and is referred to as the RL loss. The second component of the loss, $l_{cc}(\theta, \phi)$, corresponds to the loss associated with the consistency constraint and is referred to as consistency loss. $\alpha$ is a hyper-parameter to scale the consistency loss component with respect to the RL loss.

## 3.2 Observation Space Model

For the observation space models, we represent the environment as a Markov Decision Process $\mathcal{M}$ with an unknown state transition function $f : \mathcal{S} \times \mathcal{A} \to \mathcal{S}$. Given a starting state $s_t \in \mathcal{S}$, the agent learns a policy function $\pi_t$ to choose an action $a_t \in \mathcal{A}$ and a dynamics model $\hat{f}$ to predict the next state $s_{t+1}$ given a state-action pair $(s_t, a_t)$. We use the hybrid model-based and model-free (Mb-Mf) algorithm (Nagabandi et al., 2017) as the baseline to design and learn the transition function and the policy. They propose to use a trained, deep neural network based dynamics model to initialize a model free learning agent to combine the sample efficiency of model-based approaches with the high task-specific performance of model-free methods. Both the transition function and the policy are parameterized using neural networks (Gaussian Policies) as $\hat{f}_\theta(s_t, a_t)$ and $\pi_\phi(s_t)$ where $\theta$ and $\phi$ denote the parameters of the dynamics model and the policy respectively. The details about model and policy implementation are provided in the appendix 7.1.1.

In the close loop setup, the agent starts in a state $s_0$. At any time $t$, it is in state $s_t$, it chooses an action $a_t \sim \pi_t(a_t|s_t)$, receives a reward $r_t$ and observes the next state $s_{t+1}$ which it uses to choose the next action $a_{t+1}$. In the open loop setup, the agent starts in a state $s_0$. At any time $t$, it is in state $s_t$, while it imagines to be in state $s_t^I$. It chooses an $a_t \sim \pi_t(s_t)$, imagines the next state $s_{t+1}^I = \hat{f}(s_t^I, a_t)$. Simultaneously, the action $a_t$ is simulated in the environment to obtain the next environment state $s_{t+1}$. These environment states are needed to ensure consistency between the learner's imagination and actual state transitions. As described in equation 1, we encode the two state transition sequences into fixed length vectors using recurrent models and then minimize the L2 norm between them.

### 3.3 State Space Model

In case the observation space is high dimensional, as in case of pixel-wise observations(from high dimensional image data), state space models may be used to model the dynamics of the environment. These models can be computationally more efficient than the observation space models (as in pixel space) as they make predictions at a higher level of abstraction and learn a compact representation of the observation. Further, it may be easier to model the environment dynamics in the latent space as compared to the high dimensional pixel space.

We use the state-of-the-art *Learning to Query* model (Buesing et al., 2018) as our state space model. Consider a learning agent operating in an environment that produces an observation $o_t$ at every time-step $t$. These observations can be high-dimensional and highly redundant (for modelling the dynamics of the environment). The agent learns to encode these observations $(o_t)$ into compact state-space representations $(s_t)$ using an encoder $e$. The agent learns a policy function $\pi$ to choose actions $a_t \sim \pi(a_t|s_t)$.

The environment dynamics is given by an unknown observation transition function $f : \mathcal{O} \times \mathcal{A} \to \mathcal{O}$ and the agent aims to learn the model dynamics in state-space representation using a state transition function $\hat{f}$. Both the policy and state transition functions are parameterized using neural networks as $\pi_\phi$ and $\hat{f}_\theta$ where $\phi$ and $\theta$ represents the parameters of the policy and the transition function respectively. A latent variable $z_t$ is introduced per timestep to introduce stochasticity in state transition function. The observation space decoding $o_{t+1}$ can be obtained from the pixel space encoding as $o_{t+1} \sim p(o_{t+1}|s_t, z_t)$. We now describe the steps in the closed loop and open loop setup.

**Close Loop:** The agent starts in some state $s_0$ and receives an observation $o_1$ from the environment.

1. At time $t$, the agent is in a state $s_{t-1}$ and receives an observation $o_t$ from the environment.
2. $z_t \sim q(z_t|e(o_t), s_{t-1}, a_{t-1})$
3. Transition to a new state, $s_t = \hat{f}_\theta(z_t, s_{t-1}, a_{t-1})$
4. Choose an action $a_t = \pi(a_t|s_t)$
5. Decode the state $s_t$ into observation $o_{t+1} \sim p(o_{t+1}|s_t, z_t)$

**Open Loop:** The agent starts in some state $s_0$.

1. At time $t$, the agent is in a state $s_{t-1}$.
2. $z_t \sim p(z_t|s_{t-1}, a_{t-1})$
3. Transition to a new state, $s_t = \hat{f}_\theta(z_t, s_{t-1}, a_{t-1})$
4. Choose an action $a_t = \pi(a_t|s_t)$

The open loop setup for the state space models is quite similar to the case of observation space models. At time $t$, the agent is in state $s_t$, chooses an $a_t = \pi(a_t|s_t)$, transitions to the next state $s_{t+1}^I$ (using the learner's dynamics function). Simultaneously, the action $a_t$ is simulated in the external environment to obtain the next environment observation $o_{t+1}$. These environment observations are then encoded into the latent state and are needed to ensure consistency between the learner's imagined state transition and the actual state transitions in the real environment. $s_{1:T}^I$ denotes the sequence of states that the agent imagines and $o_{1:T}$ denotes the sequence of observations that the agent obtains from the environment. These observations are encoded into the state space to yield a sequence of encoded environment observations $s_{1:T}$. We want to make the behaviour of sequence $s_{1:T}$ indistinguishable from $s_{1:T}^I$. We follow the same approach as observation space models where we encode the two state transition sequences into fixed length vectors using recurrent models and then minimize the L2 norm between them (as described in equation 1). The agent is trained by imitation learning using trajectories sampled using an expert policy. The details about the model and policy implementation are provided in the appendix 7.1.2.

Since most of the latent models with stochastic dynamics are trained with one step ahead predictions, they tend to suffer from inconsistent predictions when predicting multiple time steps into the future.

On the other hand, stochasticity is also important to capture long term dependencies. By using the proposed consistency loss in the latent space, we can enforce that the multi-step predictions be grounded in the observations from the actual environment. Hence, using the proposed consistency loss to improve the long term predictions (as shown empirically) can also be seen as a regularizer.

## 4 RELATED WORK

**Model based RL** A large majority of the literature in policy search relies on model-free methods, where no prior knowledge of the environment is required to find an optimal policy, through either policy improvement (value-based methods, Rummery & Niranjan (1994); Mnih et al. (2015a)), or direct policy optimization (policy gradient methods, Mnih et al. (2016); Schulman et al. (2015a)). Although this is conceptually simple, these algorithms have a high sample complexity. To improve their sample-efficiency, one can learn a model of the environment alongside the policy, to sample experience from. PILCO (Deisenroth & Rasmussen, 2011) is a model-based method that learns a probabilistic model of the dynamics of the environment, and incorporates the uncertainty provided by the model for planning on long-term horizons.

This model of the dynamics induces a bias on the policy search though. Previous work has tried to address the model-bias issue of model-based methods, by having a way to characterize the uncertainty of the models, and by learning a more robust policy (Deisenroth & Rasmussen, 2011; Rajeswaran et al., 2016; Lim et al., 2013). Model Predictive Control (MPC, Lenz et al., 2015) has also been proposed in the literature to account for imperfect models by re-planning at each step, but it suffers from a high computational cost.

There is no sharp separation between model-free and model-based reinforcement learning, and often model-based methods are used in conjunction with model-free algorithms. One of the earliest example of this interaction is the classic Dyna algorithm (Sutton, 1991), which takes advantage of the model of the environment to generate simulated experiences, which get included in the training data of a model-free algorithm (like Q-learning, with Dyna-Q). Extensions of Dyna have been proposed (Silver et al., 2008; Sutton et al., 2012), including deep neural-networks as function approximations. Recently, the Model-assisted Bootstrapped DDPG (MA-DDPG, Kalweit & Boedecker, 2017) was proposed to incorporate model-based rollouts into a Deep Deterministic Policy Gradient method. Recently, (Weber et al., 2017) used a predictive model in Imagination-Augmented Agents to provide additional context to a policy network.

**Off-policy learning:** There are different approaches which combine on-policy learning algorithms with off-policy samples. Recent examples of this approach include the interpolated policy gradient (Gu et al.), PGQ (O'Donoghue et al., 2016) and ACER (Wang et al., 2016), which combine policy gradient learning with ideas from off-policy learning, and a methodology inspired by Q-learning. While we can incorporate experience from a behaviour policy to learn both the model of the environment as well as the policy (see Section 5.2.2), our method remains orthogonal to these works. We propose to ensure consistency between the open-loop and the closed-loop pathways as a means to learn a stronger policy, and better dynamics model. As such, our approach can be applied to a wide range of existing RL setups. Several works have incorporated auxiliary loses which results in representations which can generalize. Jaderberg et al. (2016) considered pseudo reward functions which helps to generalize effectively across different Atari games. In this work, we propose to use the consistency loss for improving the dynamics model in the context of reinforcement learning.

## 5 EXPERIMENTAL RESULTS

We designed our experiments to answer the following questions:

- How does the proposed *Consistent Dynamics* model compares against the state-of-the-art approaches for both observation space models and state space models in terms of sample complexity and asymptotic performance?
- Does adding the consistency constraint actually results in better dynamics model?

All the experiments are performed using 3 random seeds. We consider different baselines for the the observation space model and the latent space models and describe them in the subsequent sections.

## 5.1 Observation Space Models

We use the hybrid model-based and model-free (*Mb-Mf*) algorithm (Nagabandi et al., 2017) as the baseline model for the observation space models. In this setup, the policy and the dynamics model are learnt jointly. The implementation details for these models have been described in the appendix 7.1.1. We quantify the advantage of using consistency constraint by considering 4 classical Mujoco environments from RLLab (Duan et al., 2016): Ant ($S \in R^{41}$, $A \in R^8$), Humanoid ($S \in R^{142}$, $A \in R^{21}$), Half-Cheetah ($S \in R^{23}$, $A \in R^6$) and Swimmer ($S \in R^{17}$, $A \in R^3$). For computing the consistency loss, the learner's dynamics model is unrolled for $k = 20$ steps. The imagined state transitions and the actual state transitions are encoded into fixed length real vectors using GRU Cho et al. (2014). We consider the effect of changing the unrolling length $k$ as part of ablation studies.

### 5.1.1 Average Episodic Return

The average episodic return (and the average discounted episodic return) is a good estimate of the effectiveness of the jointly trained dynamics model and policy. To show that consistency constraint helps in learning a more powerful policy and a better dynamics model, we compare the average episodic rewards for the baseline *Mb-Mf* model (which does not use consistency loss) and the proposed *consistent dynamics* model (which does use the consistency loss). We expect that using consistency would either lead to higher rewards or would enable the agent to achieve the same level of rewards (as no-consistency case) but in fewer updates.

Figure 2 compares the average episodic returns for the agents trained with and without consistency. We observe that using consistency helps to learn a better policy in fewer updates for all the four environments. A similar trend is obtained for the average discounted returns (as shown in figure 8 in appendix 7.2.2). Since we are learning both the policy and the model of the environment at the same time, these results indicate that using the consistency constraint helps to jointly learn a more powerful policy and a better dynamics model.

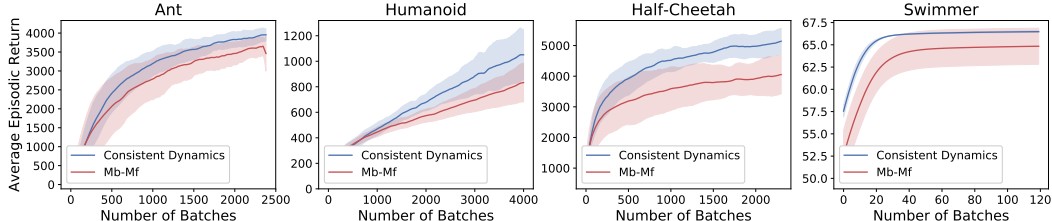

Figure 2: Comparison of the average episodic returns, for *Mb-Mf* agent and *consistent dynamics* agent on the Ant, Humanoid, Half-Cheetah and Swimmer environments (respectively). Note that the results are averaged over 100 batches for Ant, Humanoid and Half-Cheetah and 10 batches for Swimmer.

### 5.1.2 Effect of changing $k$

During the open-loop setup, the dynamics model is unrolled for $k$ timesteps. The choice of $k$ could be a critical hyper-parameter for controlling the effect of consistency constraint.

We study the effect of changing $k$ (during training) on the average episodic return for the Ant and Humanoid tasks, by training the agents with $k \in \{5, 20\}$. As an ablation, we also include the case of training the policy without using a model, in a fully model-free fashion. We would expect that a smaller value of $k$ would push the average episodic return of the *consistent dynamics* model closer to the *Mb-Mf* case. Figure 3 (Left) shows that a higher value of $k$ ($k = 20$) leads to better returns for both tasks.

## 5.2 State Space Models

State space models are useful in scenarios where the observation space is high-dimensional and possibly redundant (eg. pixels-space observations). In such cases, the state space models may be

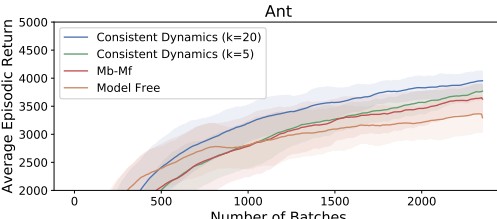 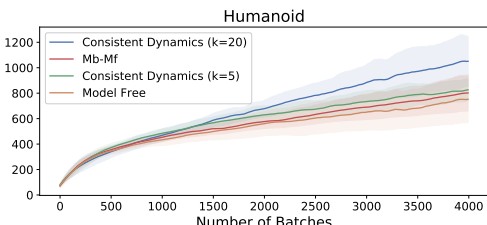

Figure 3: Average episodic return on Ant and Humanoid environments, for the agent without a model of the environment (model-free), the Mb-Mf agent without any consistency constraint, and the consistent dynamics that are trained with a consistency constraint over time horizons of length 5 and 20. Note that the results are averaged over 100 batches for Ant, Humanoid and Half-Cheetah and 10 batches for Swimmer.

used to learn the model dynamics in a condensed latent space. These setups are more challenging than the observation space setup as here the agent also needs to learn an encoder and a decoder.

We use the state-of-the-art *Learning to Query* model (Buesing et al., 2018) as the baseline state space model. We train an expert policy for sampling high-reward trajectories from the environment. The details about the training setup are described in Appendix 7.1.2. In the *Learning to Query* agent, the trajectories are used to train the policy $\pi_\phi$ using imitation learning and the dynamics model by maximum likelihood. We consider 3 continuous control tasks from the OpenAI Gym suite (Brockman et al., 2016): Half-Cheetah, Fetch-Push (Plappert et al., 2018) and Reacher. During open loop, the dynamics model is unrolled for $k = 10$ steps for Half-Cheetah and $k = 5$ for Fetch-Push and Reacher.

### 5.2.1 EVALUATING DYNAMICS MODELS

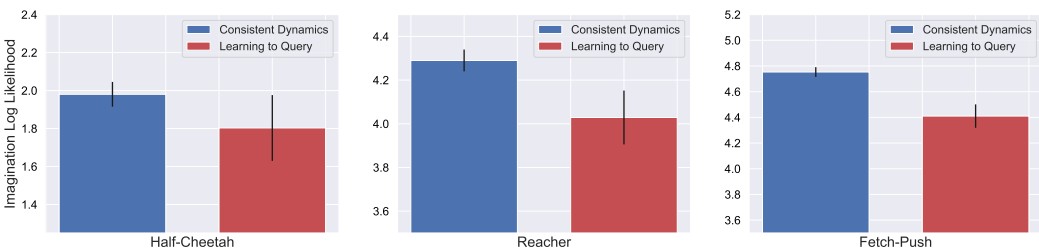

Figure 4: Comparison of the imagination log likelihood for the open loop setup for *Consistent Dynamics* agent and *Learning to Query* agent. The plots correspond to Half-Cheetah, Reacher and Fetch-Push environments (respectively). The bars represents the values corresponding to the trained agent, averaged over the last 50 batches of training. Using consistency constraint leads to a better dynamics model for all the 3 environments.

We want to show that the consistency constraint helps to learn a better dynamics model of the environment. Since we learn a dynamics model over the states, we also need to jointly learn an observation model (decoder, see appendix 7.1.2) conditioned on the states. We can then compute the log-likelihood of trajectories in the real environment (sampled with the expert policy) under this observation model. We compare the log-likelihoods corresponding to these observations for the *Learning to Query* agent (trained without the consistency loss) and *Consistent Dynamics* agent (trained with the consistency loss). We expect that the *Consistent Dynamics* agent would achieve a higher log likelihood.

Figure 4 shows that in terms of imagination log likelihood, the *Consistent Dynamics* agent outperforms the *Learning to Query* baseline agent for all the 3 environments indicating that the agent learns a more powerful dynamics model of the environment. Note that in case of Fetch-Push and

Reacher, we see improvements in the log-likelihood, even though the dynamics model is unrolled for just 5 steps.

### 5.2.2 LEARNING THE POLICY BY IMITATION LEARNING

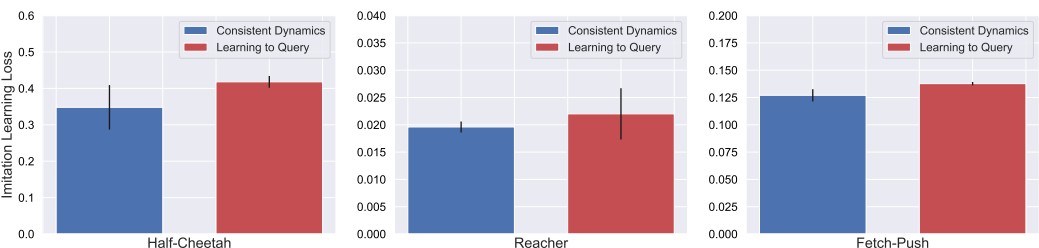

Figure 5: Comparison of the imitation learning loss for the *Consistent Dynamics* agent and *Learning to Query* agent. The plots correspond to Half-Cheetah, Reacher and Fetch-Push environments (respectively). The bars represents the values corresponding to the trained agent, averaged over the last 50 batches of training. Using consistency constraint leads to a more powerful policy.

For the state-space models, we used the expert trajectories to train our policy $\pi_\phi$ using imitation learning. To show that the consistency constraint helps to learn a more powerful policy, we compare the imitation learning loss for the *Learning to Query* agent (trained without the consistency loss) and *Consistent Dynamics* agent (trained with the consistency loss) in figure 5

## 6 CONCLUSION

In this paper, we formulate a way to ensure consistency between the predictions of a dynamics model and the real observations from the environment thus allowing the agent to learn powerful policies, as well as better dynamics models. The learning agent, in parallel, (i) builds a model of the environment and (ii) engages in an interaction with the environment. This results in two sequences of state transitions: one in the real environment where the agent actually performs actions and other in the agent's dynamics model (or the "world") where it imagines taking actions, and hallucinates the state transitions. We apply an auxiliary loss which encourages the behaviour of state transitions across the two sequences to be indistinguishable from each other. We evaluate our proposed approach for both observation space models, and state space models and show that the agent learns a more powerful policy and a better generative model. Future work would consider how these 2 interaction pathways could lead to more targeted exploration. Furthermore, having more flexibility over the length over which we unroll the model could allow the agent to take these decisions over multiple timescales.

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

## 7 APPENDIX

### 7.1 ENVIRONMENT MODEL

#### 7.1.1 OBSERVATION SPACE MODEL

We use the experimental setup, environments and the hybrid model-based and model-free (Mb-Mf) algorithm as described in (Nagabandi et al., 2017)[1]. We consider two training scenarios: training a model-based learning agent with and without the consistency constraint. The consistency constraint is applied by unrolling the model for multiple steps using the observations predicted by the learner's dynamics model (closed-loop setup). We train an on-policy RL algorithm for Cheetah, Humanoid, Ant and Swimmer tasks from RLLab (Duan et al., 2016) control suite. We report both the average discounted and average un-discounted reward obtained by the learner in the two cases: with and without the use of consistency constraint. The model and policy architectures for the observation space models are as follows:

1. *Transition Model*: The transition model $\hat{f}_\theta(s_t, a_t)$ has a Gaussian distribution with diagonal covariance, where the mean and covariance are parametrized by MLPs (Schulman et al., 2015a), which maps an observation vector $s_t$ and an action vector $a_t$ to a vector $\mu$ which specifies a distribution over observation space. During training, the log likelihood $p(s|\mu)$ is maximized and state-representations can be sampled from $p(s|\mu)$.
2. *Policy*: The learner's policy $\hat{\pi}_\phi(s_t)$ is also a Gaussian MLP which maps an observation vector $s$ to a vector $\mu_{policy}$ which specifies a distribution over action space. Like before, the log-likelihood $p(a|\mu)$ is maximized and actions can be sampled from $p(a|\mu)$.

Learner's policy and the dynamics model are implemented as Gaussian policies with MLPs as function approximations, and are trained using TRPO (Schulman et al., 2015a). Following the hybrid Mb-Mf approach(Nagabandi et al., 2017), we normalize the states and actions. The dynamics model is trained to predict the change in state $\Delta s_t$ as it can be difficult to learn the state transition function when the states $s_t$ and $s_{t+1}$ are very similar and the action $a_t$ has a small effect on the output.

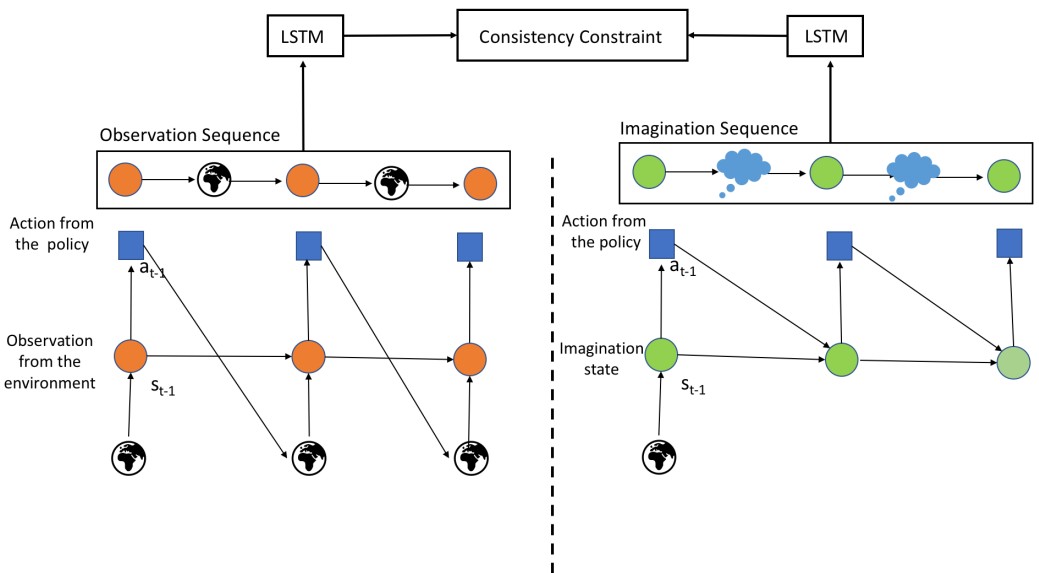

Figure 6: Open-loop and closed-loop pathways in the Observation Space Models. The consistency constraint aims to make the behaviour of the open loop predictions indistinguishable from the close loop behaviour

---

[1]Code available here: https://github.com/nagaban2/nn_dynamics

### 7.1.2 STATE SPACE MODEL

We use the state-of-the-art *Learning to Query* model (Buesing et al., 2018) as our state space model. The model and policy architecture for the state space models is as follows:

1. *Encoder*: The learner encodes the pixel-space observations ($64 \times 64 \times 3$) from the environment into state-space observations (256 dimensional vectors) with a convolutional encoder (4 convolutional layers with $4 \times 4$ kernels, stride 2 and 64 channels). To model the velocity information, a stack of the latest 4 frames is used as the observation. The pixel-space observation at time $t - 1$ is denoted as $o_{t-1}$, and is encoded into state-space observation $s_{t-1}$.

2. *Transition Model*: The transition model is a Long Short-Term Memory model (LSTM, Hochreiter & Schmidhuber, 1997), that predicts the transitions in the state space. For every time-step $t$, latent variables $z_t$ are introduced, whose distribution is a function of previous state-space observation $s_{t-1}$ and previous action $a_{t-1}$. ie $z_t \sim p(z_t|s_{t-1}, a_{t-1})$. The output of the transition model is then a deterministic function of $z_t, s_{t-1}$, and $a_{t-1}$. ie $s_t = f(z_t, s_{t-1}, a_{t-1})$.

3. *Stochastic Decoder*: The learner can decode the state-space observations back into the pixel-space observations by use of stochastic convolutional decoder. The decoder takes as input the current state-space observation $s_t$ and the current latent variable $z_t$ and generates the current observation-space distribution from which the learner could sample an observation. ie $o_{t+1} \sim p(o_{t+1}|s_t, z_t)$. This observation model is Gaussian, with a diagonal covariance.

In the closed-loop trajectory, when the learner cannot interact with the environment, the latent variables are sampled from the prior distribution $p(z_t|s_{t-1}, a_{t-1})$. The latent variables are sampled from Normal distributions with diagonal covariance matrices. Since we cannot compute the log-likelihood $L(\theta)$ in a close form for the latent variable models, we minimize the evidence lower bound $\text{ELBO}(p_{posterior}) \leq L(\theta)$. As discussed previously, the consistency constraint is applied between the open-loop and closed-loop predictions with the aim of making their behaviour as similar as possible. Figure 7 shows a graphical representation of the open-loop and close-loop pathways in the state-space model.

**Expert policy**   Having access to some policy trained on a large number of experience is required to sample high-quality trajectories with pixel-observations. To train these expert policies, we used policy-based methods such as Proximal Policy Optimization (PPO, Schulman et al., 2017) for the half-cheetah and reacher environments, or Deep Deterministic Policy Gradient with Hindsight Experience Replay (DDPG with HER, Andrychowicz et al., 2017) for the pushing task. The architectures and hyper-parameters used are similar to the ones given by the Baselines library (Dhariwal et al., 2017). Note that these expert policies were trained on the state representation of the agents (ie. the positions and velocities of their joints), while the trajectories were generated with pixel-observations captured from a view external to the agent.

## 7.2 RESULTS

### 7.2.1 OBSERVATION SPACE MODELS

### 7.2.2 STATE SPACE MODELS

## 7.3 ROBUSTNESS TO COMPOUNDING ERRORS

In this section, we investigate the robustness of the proposed approach in terms of compounding errors. When we use the recurrent dynamics model for prediction, the ground-truth sequence is not available for conditioning. This leads to problems during sampling as even small prediction errors can compound when sampling for a large number of steps. We evaluate the proposed model for robustness by predicting the future for much longer timesteps (50 timesteps) than it was train on (10 timesteps). More generally, in figure 9, we demonstrate that this auxiliary cost helps to learn a better model with improved long-term dependencies by using a training objective that is not solely focused on predicting the next observation, one step at a time.

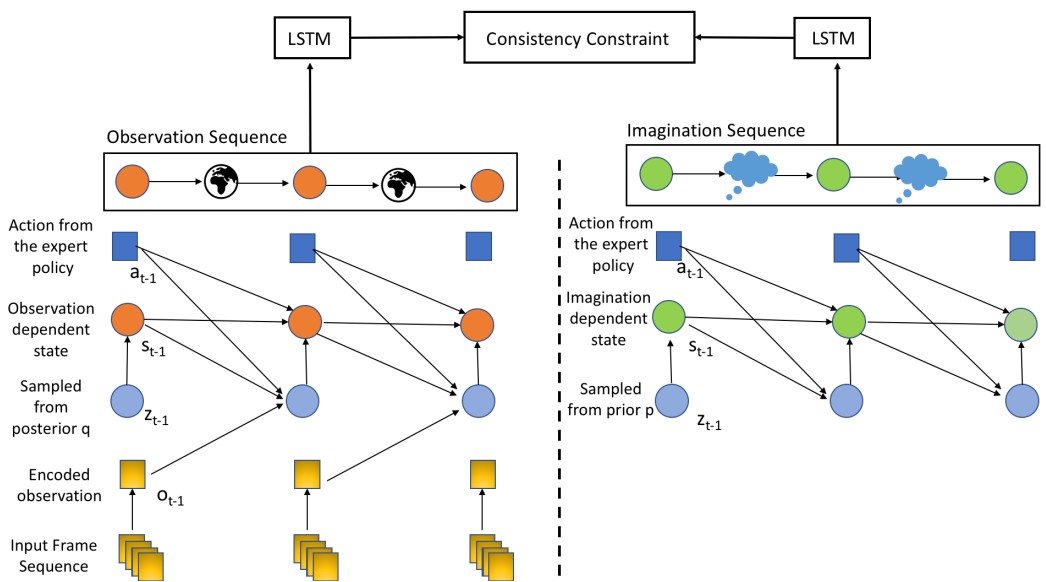

Figure 7: Open-loop and closed-loop pathways in the State Space Models. The consistency constraint aims to make the behaviour of the open loop predictions indistinguishable from the close loop behaviour

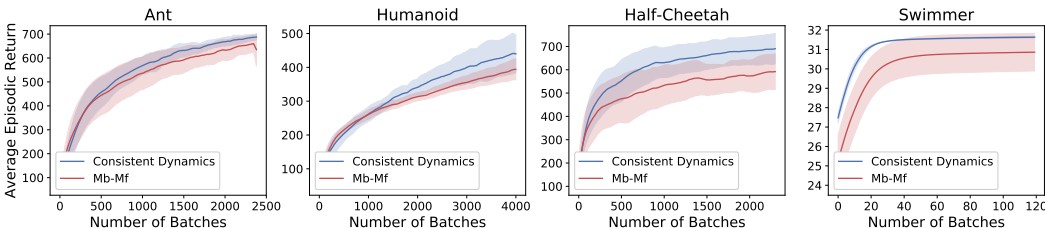

Figure 8: Comparison of the average episodic discounted rewards, for agents trained with and without consistency for the Ant, Humanoid, Half-Cheetah and Swimmer environments (respectively). Using consistency constraint leads to better rewards in a fewer number of updates for all the cases. Vertical lines in the rightmost figure show the points of saturation with equal return. Note that the results are averaged over 100 batches for Ant, Humanoid and Half-Cheetah and 10 batches for Swimmer.

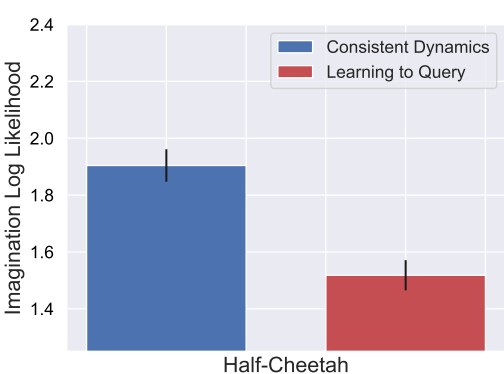

Figure 9: Comparison of the imagination log likelihood for the *Consistent Dynamics* agent and *Learning to Query* agent for Half-Cheetah. The agents were trained with sequence length of 10 but during testing, the dynamics models were evaluated for length 50. The bars represents the values corresponding to the trained agent, averaged over the last 50 batches of training. Using consistency constraint leads to an improved dynamics model (as it achieves better log-likelihood)

