# OpenReview forum: "Learning powerful policies and better dynamics models by encouraging consistency"
_ICLR.cc/2019/Conference_

### Official Review · AnonReviewer2 · 2018-11-02
**Needs clarification.**

**Rating:** 5
**Confidence:** 3

**Review:**


-------------
Summary
-------------
The authors propose to train a policy while concurrently learning a dynamics model. In particular, the policy is updated using both the RL loss (rewards from the environment) and the "consistency constraint", which the authors introduce. This consistency constraint is a supervised learning signal, which compares trajectories in the environment with trajectories in the imagined world (produced with the dynamics model).

---------------------
Main Feedback
---------------------
I feel like there might be some interesting ideas in this work, and the results suggest that this approach performs well. However, I had a difficult time understanding how exactly the method works, and what its advantages are. These are my main questions:

1) At the beginning of Section 4 the authors write "The learning agent has two pathways for improving its behaviour: (...) (ii) the open loop path, where it imagines taking actions and hallucinates the state transitions that could happen". Do you actually do this? This is not mentioned in anywhere. And as far as I understand, the reward function is not learned - hence there will be no training signal in the open loop path. Does the reward signal always come from the true environment?
2) Is the dynamics model used for anything else than action-selection during training? Planning? If not, I don't really understand the results and why this works at all (k=20 being better than k=5, for example).
3) Is the dynamics model pre-trained in any way? I find it surprising that the model-free method and the proposed method perform similar at the beginning (Figure 3). If the agent chooses its actions based on the state that is predicted by the dynamics model, this should throw off the learning of the policy at the beginning (when the dynamics model hasn't learned anything sensible yet).

-----------------------
Other Questions
-----------------------
4) How exactly does training without the consistency constraint look? Is this the same as k=1?
5) Could the authors comment on the evaluation protocol in the experimental section? Are the results averages over multiple runs? If so, it would help to see confidence intervals to make a fair assessment of the results.
6) For the swimmer in Figure 2, the two lines (with consistency and without consistency) start at different initial returns, why is that so? If the same architecture and seed was used, shouldn't this be the same (or can you just not see it in the graph)?

---------
Clarity
---------
The title and introduction initially gave me a slightly wrong impression on what the paper is going to be about, and several things were not followed up on later in the paper.
Title:
8) "generative models" reminds of things like a VAE or GAN; however, I believe the authors mean "dynamics models" instead
9) "by interaction" is a bit vague as to what the contribution is (aren't policies and dynamic models in general trained by interacting with the environment?); the main idea of the paper is the consistency constraint
Abstract / Introduction:
10) The authors talk about humans carrying out "experiments via interaction" to help uncover "true causal relationships". This idea is not brought up again in the methods section, and I don't see evidence that with the proposed approach, the policy does targeted experiments to uncover causal relationships. It is not clear to me why this is the intuition that motivates the consistency constraint.
11) As the authors state in the introduction, the hope of model-based RL is better sample complexity. This is usually achieved by using the model in some way, for example by planning several steps ahead when choosing the current action. Could the authors comment on where they would place their proposed method - how does it address sample complexity?
12) In the introduction, the authors discuss the problem of compounding errors. These must be a problem in the proposed method as well, especially as k grows. Could the authors comment on that? How come that the performance is so good for k=20?
13) The authors write that in most model-based approaches, the dynamics model is "learned with supervised learning techniques, i.e., just by observing the data" and not via interaction. There's two things I don't understand: (1) in the existing model-based approaches the authors refer to, the policy also interacts with the world to get the data to do supervised learning - what exactly is the difference? (2) The auxiliary loss "which explicitly seeks to match the generative behaviour to the observed behaviour" is just a supervised learning loss as well, so how is this different?

For me, it would help the readability and understanding of the paper if some concepts were introduced more formally.
14) In Section 2, it would help me to see a formal definition of the MDP and what exactly is optimised. The authors write "optimise a reward signal" and "maximise its expected reward", however I believe it should be the expected cumulative reward (i.e., return).
15) The loss function for the dynamics model is not explicitly stated. From the text I assume that it is the mean squared error for the per-step loss, and a GAN loss for the trajectory-wise loss.
16) Could the authors explicitly state what the overall loss function is, and how the RL and supervised objective are combined? Is the dynamics model f trained only on the supervised loss, and the policy pi only on the RL loss?
17) In 2.3 the variable z_t is not formally introduced. What does it represent?

------------------------
Other Comments
------------------------
18) I find it problematic to use words such as "hallucination" and "imagination" when talking about learning algorithms. I would much prefer to see formal/factual language (like saying that the dynamics model is used to do make predictions / do planning, rather than that the agent is hallucinating).

-- edit (19.11.) ---
- updated score to 5
- corrected summary

---

> ### Author Response · Authors · 2018-11-15
> **Response to Reviewer 2 - Part 1**
>
> We thank the reviewer for such a detailed feedback. We have conducted additional experiments to address the concerns raised about the evaluation, and we clarify specific points below. We believe that these additions address all of your concerns about the work, though we would appreciate any additional comments or feedback that you might have. We acknowledge that the paper was certainly lacking polish and accept that this may have made the paper difficult to read in places. We have uploaded a revised version in which we have revised the problem statement and writing as per the reviewer's suggestions. We briefly summarize the key idea of the paper and then address the specific concerns.
>
> What is the idea?
> ========
>
> Our goal is to provide a mechanism for the agent to lean a better dynamics model as well as more powerful policy by ensuring the consistency in their predictions (such that predictions from the model are grounded in the real environment).
>
> This mechanism enables the agent to have a direct “interaction” b/w the agent’s policy and its dynamics model. This interaction is different from the standard approaches in reinforcement learning where the agent uses the policy to sample trajectories over which the agent is trained, and then use these sampled trajectories to learn the dynamics model. In those cases, there is no (direct) mechanism for the dynamics model to affect the policy, and hence there is no “direct interaction” between the policy and the dynamics model.  In our case, both the policy and the model are trained jointly while making sure that the predictions from the dynamics model are consistent with the observation from the environment. This provides a mechanism where learning a model can itself change the policy (thus “interacting” with the environment) instead of just training on the data coming from a policy which is trained independently of how well the model performs on the collected.
>
> A practical instantiation of this idea is the consistency loss where we ensure consistency between the predictions from the dynamics model and the actual observations from the environment and this simple baseline works surprisingly well compared to the state of the art methods (as demonstrated by our experiments) and that others have not tried it before. Applying consistency constraint means we have two learning signals for the policy: The one from the reinforcement learning loss (i.e maximize return) and the other due to consistency constraint. We show that adding the proposed consistency constraint helps the agent to learn better dynamics model and as well as better policy for both observation space models and state space models. We compare against strong baselines:
>
> Hybrid model-based and model-free (Mb-Mf) algorithm (Nagabandi et al [1])
> Learning and Querying Fast Generative Models for Reinforcement Learning (Buesing et al [2]) - This is a state-of-the-art model for state space models.
>
> Our evaluation protocol considers a total of 7 environments and we show that using the consistency constraint leads to better generative models (in terms of log likelihood) and more powerful policy (average return) for all the cases. All the experiments are averaged over 3 random seeds and are plotted with 1 standard deviation interval.
>
> Our key contribution is the proposal of using the consistency loss which helps to learn more powerful policy and better dynamics model (as demonstrated over different tasks) while being very easy to integrate with existing model-based RL approaches. While our method is relatively simple, we are not aware of prior works that show something similar, and we believe such a simple baseline would be useful for anyone who’s working on model-based RL. Further, our experiment demonstrates the effectiveness of the approach. If we are mistaken regarding prior works, please let us know!
>
> We would like to emphasize that our work presents an extensive comparative evaluation, and we believe that these results should be taken into consideration in evaluating our work. We compare multiple approaches across more than 5 simulated tasks to the state of the art methods. Hopefully, our clarifications are convincing in terms of explaining why the evaluation is fair and rigorous, and we would, of course, be happy to modify it as needed. But at a higher level, the fact that such simple model-based approaches work better than somewhat complex model-free approaches actually is the point of the paper to me.
>
> Continued...

---

> > ### Comment · AnonReviewer2 · 2018-11-19
> > **Reply**
> >
> > Thank you for the detailed answer. I seem to have misunderstood parts of the paper. In particular, the section in the author’s answer on “Why is different from just learning a model based on k-step prediction?” clarified that the policy is updated using two learning signals - the RL loss, and the loss from the consistency constraint. This is the key for why the method works (and now the results make much more sense to me). The dynamics model is not used for action selection, but only as an additional learning signal for the policy. Learning a good dynamics model is a nice side product, but this model is not used at test time.
> >
> > Kudos to the authors for being so open for feedback, and updating the paper title and adding more formal explanations of the concepts discussed in the paper.
> >
> > Overall, I think the idea is simple but works well, and the paper is much improved. Hence I am willing to increase my score to 5.
> >
> > It’s difficult for me to assess how significant the idea is - the authors say their results are state of the art, but the other reviewers say that it lacks important comparisons, and I am not well versed enough in the literature to make an informed evaluation on this point. While the main intuition is clearer to me now, I feel like the introduction is still a little convoluted (e.g., again my point that in the introduction, the authors motivate that the agent tries to uncover causal relationships in the environment - but this is not followed up on in the rest of the paper).
> >
> > In general, I am not sure if introducing an auxiliary loss that boosts performance on a selected group of MuJoCo environments warrants acceptance, if the authors cannot give more insight and analysis for why this works so well (and in which cases it might actually not work well?). Such insight (both the intuition of the authors, and in experiments) would be very interesting for other researchers that might want to use this auxiliary loss.
> >
> > Notes:
> > - It might be interesting to discuss connections to other works in RL that use auxiliary losses, e.g. “Reinforcement Learning with Unsupervised Auxiliary Tasks” (Jaderberg et al. 2016)
> > - Sec 2., write “discounted sum of rewards” instead of “sum of rewards”.
> > - Sec 2 uses $/phi$ for the learned model, Sec 3 uses $\theta$ for the model and $\phi$ for the policy - please make this consistent. It might help to already introduce the parameterised policy in Sec 2 as well.
> > - Sec 3.1, last paragraph, should be “expected return” instead of “expected reward”

---

> > > ### Author Response · Authors · 2018-11-19
> > > **Thanks for reply**
> > >
> > > We appreciate that the reviewer took time to read our lengthy rebuttal, and increased their score. :)
> > >
> > > "It’s difficult for me to assess how significant the idea is"
> > >
> > > It's a simple addition which improves the quality of the dynamics model, as well as the policy. Now, we acknowledge that the paper was certainly lacking polish and accept that this may have made the paper difficult to read in places (for you as well as for other reviewers). And hence it might have been confusing for the other reviewers as well. Otherwise, we compare (and outperform) the proposed method to state of the art methods like MB-MF, dyna and Learning to query. (As the Reviewer 1 has suggested ONLY these baselines too).
> > >
> > > [1] MB-MF https://arxiv.org/abs/1708.02596
> > > [2] Dyna, https://www.sciencedirect.com/science/article/pii/B9781558601413500304
> > > [3] Learning to query https://arxiv.org/abs/1802.03006
> > >
> > >
> > > "I am not sure if introducing an auxiliary loss that boosts performance on a selected group of MuJoCo environments warrants acceptance,"
> > >
> > > We believe that comparing to [1] provides a good baseline.  Out of all these [1], [2] tested the ideas on Mujoco envs, whereas [3] tested the env on Atari games where they first pretrain the dynamics model, and then use that model.  We compared against  [3] on a challenging Mujoco task, where we learn the model directly from the *pixels*. We also note that [3] does not have an open-source implementation, and they ([3]) only evaluated on few atari games using millions of samples per game. We believe that comparing to such a strong baseline is very important and hence we compared to this on a challenging image based mujoco env. Since, one cant infer the velocity of the cheetah just using a particular frame, and hence learning from images make this task a partially observable task, and more challenging.
> > >
> > > " if the authors cannot give more insight and analysis for why this works so well (and in which cases it might actually not work well?)"
> > >
> > > We think that having an an auxiliary cost could always improve the performance. As We evaluate it on a large number of very different domains (when learning the model directly from pixels as well as learning the model using the state representation directly)  and find that in all cases it improves performance. For some problems, learning a model of the environment is difficult so those problems would be hard as well. This applies to any complex environment and especially partially observed ones. Our method would  help the most when the dynamics are relatively simple but the problems are still relatively hard.
> > >
> > > "if the authors cannot give more insight and analysis for why this works so well"
> > >
> > > We would be happy to add comparison to any another baseline which the reviewer has in mind.  We want to make sure, that we can do everything possible to make sure that the researchers in the future would try against such a simple baseline.

---

> ### Author Response · Authors · 2018-11-15
> **Response to Reviewer 2 - Part 2**
>
>
>
>
> Why is different from just learning a model based on k-step prediction?
> ========
>
> Our approach is different from just learning a k-step prediction model as in our case, the agent’s behavior (i.e the agent's policy) is dependent on its internal model too. In the standard case, the policy is optimized only using the RL gradient i.e maximizing expected reward and the state transition pairs (collected as the agent acts in the environment) become the supervising dataset for learning the model, and hence the policy is not affected when the model is being updated and there is no feedback from the model learning process to the policy. Hence, the data used for training the model is coming from a policy which is trained independently of how well the model performs on the collected trajectories. So the process of learning the model has no control over what kind of data is produced for its training.
>
> We propose to train both the policy and the model during the open loop. Hence the k-step predictions are used for training both the model and the policy simultaneously. Training the policy on both the RL loss and the consistency loss provides a mechanism where learning a model, can itself change the policy thus leading to a much closer interplay between the dynamics model and the policy. We show that this relatively simple approach leads to much better performance when compared to very strong baselines for both observation space and state space models for all the 7 environments we considered.
>
>
> What are the empirical results?
> ========
>
> Our evaluation protocol consists of 7 environments (Ant, Half Cheetah, Humanoid etc) and both observation space and state space models. Solving Half Cheetah environment, when observations are in the pixel space (images), is very challenging as useful information like velocity is not available.
>
> For the observation space model, we use the “Hybrid model-based and model-free (Mb-Mf) algorithm” (Nagabandi et al [1]). It is a strong baseline where the authors proposed to use a trained, deep neural network based dynamics model to initialize a model-free learning agent to combine the sample efficiency of model-based approaches with the high task-specific performance of model-free methods. For the state space models, we use the “Learning and Querying Fast Generative Models for Reinforcement Learning” (Buesing et al [2])  as the baseline. This is a state-of-the-art model for state space models. As shown by our experiments (section 5), by having this consistency constraint we outperform both these baselines.
>
> We focus on evaluating the agent for both dynamics models (in terms of imagination log likelihood, figure 4) and policy (in terms of average episodic returns and loss, figure 2, 3, 5). We show that adding the consistency constraint to the baseline models results in improvements to both the dynamics models and the policy for all the environments that we consider. All the experiments are averaged over 3 random seeds and are plotted with 1 standard deviation interval.
>
> ===============================================================================================
>
> We now refer to the specific aspects of the reviews.
>
>
> "The authors ... as an input.
>
> There seems to be a small discrepancy in the summary. The actions are always selected using the true state of the environment. When the agent is performing the open-loop, the agent transitions from one “imagined” state to another “imagined” state, unlike the closed loop state where the agent transitions between actually observed states (coming from the environment). The consistency loss ensures that the sequence of imagined states behaves similarly to the sequence of actually observed states. This aspect has been clarified in the paper in section 3.1 which talks about consistency constraint in general and describes how the consistency loss is to be computed (eq 1). Section 3.2 and section 3.3 go into the specific cases of observation space models and state space models respectively.
>
> ==============================
>
> "1) At the beginning ... environment?"
>
> Thank you for pointing out this out. We have improved the writing in the paper to make it more explicit (equation 1). We briefly summarise this aspect here for completion:
>
> Let us say that at time t, the agent is in some state s_t while it “imagines” to be in state s_t^I. In the closed loop, it samples an action a_t using the policy \pi and transitions to a new state s_{t+1} by performing the action in the environment. In the open loop, the agent performs the action a_t in its dynamic model and transitions from state s_t^I to s_{t+1} ^ I .
>
> For the closed loop, the loss comes from the reward signal. For the open loop, the loss comes in form of consistency constraint imposed on the sequence of actual state transitions and the predicted state transitions. This is described by equation 1 in section 3.1. During the open loop, both the policy and the model are updated using the consistency loss.
>
> Cont..

---

> ### Author Response · Authors · 2018-11-15
> **Response to Reviewer 2 - Part 3**
>
> "2) Is the dynamics ... for example)"
>
>
> The dynamics model is indeed being used like in other model-based approaches. k=20 works better than k=10 because now the model’s predictions are being grounded in “real observations” for a much longer time span.
>
> "3) Is the dynamics ... sensible yet)"
>
> We clarify that the agent is not using the dynamics model for action selection. The role of the dynamics model is the following - The policy is trained using both the RL loss as well as the loss from the dynamics model.
>
>
> "4) How exactly ... k=1?"
>
> The case of training without the consistency loss is the standard reward-based training of RL agents, without any consistency constraint. K=1 would correspond to the case where the consistency loss is applied on per step predictions.
>
> "5) Could the ... results."
>
> We have updated the paper to improve the experimental section  - both in terms of description of baselines and in terms of the evaluation protocol. Further,  Section 3.1 describes the different loss components and how the consistency constraint can be applied in the general. Section 3.2 and 3.3 describes the baselines and how these baselines were modified to support the consistency constraint for the observation space and the state space models respectively. All the experiments are averaged over 3 random seeds (along with 1 standard deviation interval) are plotted.
>
> We summarize the baselines and the evaluation protocol here:
>
> Our evaluation protocol consists of 7 environments (Ant, Half Cheetah, Humanoid etc) and both observation space and state space models. For the observation space model, we use the “Hybrid model-based and model-free (Mb-Mf) algorithm” (Nagabandi et al [1])  which is a very strong baseline and for the state space models, we use the “Learning and Querying Fast Generative Models for Reinforcement Learning” (Buesing et al [2])  as the baseline. This model is a state-of-the-art model for state space models.
>
> We focus on evaluating the agent for both dynamics models (in terms of imagination log likelihood) and policy (in terms of average episodic returns and loss). We show that adding the consistency constraint to the baseline models results in improvements to both the dynamics models and the policy for all the environments that we consider. All the experiments are averaged over 3 random seeds and are plotted with 1 standard deviation interval.
>
> Our key contribution is the proposal of using the consistency loss which helps to learn more powerful policy and better dynamics model (as demonstrated over different tasks) while being very easy to integrate with existing model-based RL approaches. While the proposed approach looks relatively simple, we are not aware of work in RL which describes and validates the benefits of imposing the consistency constraint.
>
>
> "For the ... it in the graph)?""
>
>
> We believe that the reason is the swimmer plot is averaged over 10 batches. We have added this information in the caption of the plot.
>
>
> "8) ... instead"
>
> We agree that the title could sound a little misleading. Based on the suggestion, we have updated the title to “Learning powerful policies and better dynamics models by encouraging consistency”
>
> "9) by interaction ... consistency constraint"
>
> We acknowledge that the use of “by interaction” sounds a little vague and have incorporated this feedback into the draft.
>
> "10) The authors ... constraint"
>
> Our broad goal is to provide a mechanism for the agent to interact with the environment while it is learning the dynamics model as this could be helpful in learning a more powerful policy and better dynamics model. We discuss several possible manifestations of this idea in the introduction/motivation and focus on one specific instantiation - ensuring consistency between the predictions from the dynamics model and the actual observations from the environment. We show that adding the proposed consistency constraint helps the agent to learn better dynamics model and better policy for both observation space models and state space models. It is both interesting and surprising to see that our proposed approach improves over the state of the art results despite being relatively simple thus highlighting the usefulness of the ‘interaction” with the environment.
>
> Continue

---

> ### Author Response · Authors · 2018-11-15
> **Response to Reviewer 2 - Part 4**
>
> "11) As the ... sample complexity?"
>
> We address this issue from two perspectives:
>
> Qualitatively - We propose to train both the policy and the model during the open loop. Hence the k-step predictions are used for training both the model and the policy simultaneously. Training the policy on both the RL loss and the consistency loss provides a mechanism where learning a model, can itself change the policy thus leading to a much closer interplay between the dynamics model and the policy. This approach is different from other works focusing on learning k-step prediction models. In those cases, the policy is learned solely focussing on the reward structure and the state transition trajectories (collected as the agent acts in the environment) become the supervising dataset for learning the model. There is no feedback from the model learning process to the policy learning process. So the process of learning the model has no control over what kind of data is produced (by the policy)  for its training.
>
>
> Empirical Evaluation - We show that this relatively simple approach improves the performance for both the dynamics model and the policy when compared to very strong baselines for both observation space and state space models for all the 7 environments we considered. Our evaluation protocol consists of 7 environments (Ant, Half Cheetah, Humanoid etc) and both observation space and state space models. For the observation space model, we use the “Hybrid model-based and model-free (Mb-Mf) algorithm” (Nagabandi et al [1])  which is a very strong baseline and for the state space models, we use the “Learning and Querying Fast Generative Models for Reinforcement Learning” (Buesing et al [2])  as the baseline. This model is a state-of-the-art model for state space models.
>
>
>
> "12) In the ... k=20?"
>
> This comment refers to figure 3. Here the proposed agents are trained with 2 different values of k, that is 5 and 20. Since the agent with k=20 is trained for longer sequences, it performs better than the other agent.
>
>
> "13) The authors ...different?"
>
> The key difference between our approach and existing approaches for learning the dynamics model is that in our case, the process of learning the model can change the policy.
>
> In the standard cases, the policy is learned solely focussing on the reward structure and the state transition trajectories (collected as the agent acts in the environment) become the supervising dataset for learning the model. In that setup, the policy is not updated when the model is being updated and there is no feedback from the model learning process to the policy learning process. Hence, the data used for training the model is coming from a policy which is trained independently of how well the model performs on the collected trajectories. So the process of learning the model has no control over what kind of data is produced for its training. This is what we mean by “learning the dynamics model by just observing the data”.
>
> We propose to train both the policy and the model during the open loop. Hence the k-step predictions are used for training both the model and the policy simultaneously. Training the policy on both the RL loss and the consistency loss provides a mechanism where learning a model, can itself change the policy. This is what we mean by “learning the dynamics model via interaction”. This close interplay between the dynamics model and the policy provides a pathway to the model to interact with the environment instead of just using the sampled trajectories. The resulting consistency loss helps to learn more powerful policy and better dynamics model (as demonstrated over different tasks) while being very easy to integrate with existing model-based RL approaches. It is important to note that our proposed approach improves over the state of the art results despite being relatively simple. We are not aware of work in RL which describes and validates the benefits of imposing the consistency constraint and would be happy to include references to such work.
>
>
>
> "14) In Section ... return)"
>
> We apologize for the mistake. Thanks for pointing it out. We are indeed optimizing the expected return. We have also updated section 2 to describe the MDP and the related terms in a formal manner.
>
>
> "15) The loss ... trajectory-wise loss"
>
> We have improved the section on consistency constraint (Section 3.1) to describe the consistency loss in detail. We do not use any stepwise loss. A recurrent model is used to encode the trajectory into a fixed-sized vector and the l2 loss is applied between the encoding for the trajectory of observed states and the imagined states. This has been formalized in equation 1.  Further, Section 3.2 and 3.3 describe how to modify the baselines to support the consistency constraint for observation space and state space models respectively.
>
> Continue

---

> ### Author Response · Authors · 2018-11-15
> **Response to Reviewer 2 - Part 5**
>
> "16) Could the ... RL loss?"
>
>
> We have updated the section on consistency constraint (3.1) to include an equation describing the different components of the loss function.  Both the dynamics model and the policy pi are trained on the total loss (which is a combination of the RL loss and the consistency loss)
>
> "17) In 2.3 ... represent?"
>
> We have updated the relevant section to define z_t. It refers to the latent variable introduced per timestep to introduce stochasticity in state transition function.
>
> "18) I find ... hallucinating."
>
> We have addressed this point by replacing the word hallucination with “imagination” and “prediction” as per the context.
>
>
> We would appreciate it if the reviewer could take another look at our changes and additional results, and let us know if the reviewer has request for additional changes that would alleviate the reviewer's concerns.
>
> [1]: Neural Network Dynamics for Model-Based Deep Reinforcement Learning with Model-Free Fine-Tuning - https://arxiv.org/pdf/1708.02596.pdf
>
> [2]: Learning and Querying Fast Generative Models for Reinforcement Learning - https://arxiv.org/pdf/1802.03006.pdf

---

> ### Author Response · Authors · 2018-11-20
> **Added results to evaluate the robustness of the model**
>
> Dear Reviewer
>
> We have added new evaluation results to investigate the robustness of the proposed approach in terms of compounding errors. When we use the recurrent dynamics model for prediction, the ground-truth sequence is not available for conditioning. This leads to problems during sampling as even small prediction errors can compound when sampling for a large number of steps.  We evaluate the proposed model for robustness by predicting the future for much longer timesteps (50 timesteps) than it was trained on (10 timesteps). More generally, in figure 9 (section 7.3 in appendix), we demonstrate that this auxiliary cost helps to learn a better model with improved long-term dependencies by using a training objective that is not solely focused on predicting the next observation, one step at a time.
>
> Thank you for your time! The authors appreciate the time reviewers have taken for providing feedback. which resulted in improving the presentation of our paper. Hence,  we would appreciate it if the reviewers could take a look at our changes and additional results, and let us know if they would like to either revise their rating of the paper or request additional changes that would alleviate their concerns.

---

### Official Review · AnonReviewer3 · 2018-11-05
**This paper presents the idea of learning models of the environment while interacting with it, in the form of performing the usual model-based or model-free reinforcement learning, while enforcing consistency between the real world (observations) and the model. The presented motivation is that agents, like people, can benefit through not just observing the environment and learning from it, but also by experimenting---trying actions specifically for learning**

**Rating:** 2
**Confidence:** 4

**Review:**

---Below is based on the original paper---
This paper presents a framework that allows the agent to learn from its observations, but never follows through on the motivation of experimentation---taking actions mainly for the purpose of learning an improved dynamics model. All of their experiments merely take actions that are best according to the usual model-based or model-free methods, and show that their consistency constraint allows them to learn a better dynamics model, which is not at all surprising. They do not even allow for the type of experimentation that has been done in reinforcement learning for as long as it has been around, which is to allow exploration by artificially increasing the reward for the first few times that each state is visited. That would be a good baseline against which to compare their method.

Overall:
Pros:
1. Clear writing
2. Good motivation description.

Cons:
1. Failed to connect presented work with the motivation.
2. No comparison against known methods for exploration.


----Below is based on the revision---

Thanks to the reviewers for making the paper much clearer. I have no particular issues on the items that are in the paper. However, subsections 7.2.1 and 7.2.2 are missing.

---

> ### Author Response · Authors · 2018-11-15
> **Response to Reviewer 3 - Part 1**
>
> We thank the reviewer for the feedback. We have conducted additional experiments to address the concerns raised about the evaluation, and we clarify specific points below. We believe that these additions address all of your concerns about the work, though we would appreciate any additional comments or feedback that you might have. We acknowledge that the paper was certainly lacking polish and accept that this may have made the paper difficult to read in places. We have uploaded a revised version in which we have revised the problem statement and writing as per the reviewer's suggestions. We briefly summarize the key idea of the paper and then address the specific concerns.
>
> What is the idea?
> ========
>
> Our goal is to provide a mechanism for the agent to lean a better dynamics model as well as more powerful policy by ensuring the consistency in their predictions (such that predictions from the model are grounded in the real environment).
>
> This mechanism enables the agent to have a direct “interaction” b/w the agent’s policy and its dynamics model. This interaction is different from the standard approaches in reinforcement learning where the agent uses the policy to sample trajectories over which the agent is trained, and then use these sampled trajectories to learn the dynamics model. In those cases, there is no (direct) mechanism for the dynamics model to affect the policy, and hence there is no “direct interaction” between the policy and the dynamics model.  In our case, both the policy and the model are trained jointly while making sure that the predictions from the dynamics model are consistent with the observation from the environment. This provides a mechanism where learning a model can itself change the policy (thus “interacting” with the environment) instead of just training on the data coming from a policy which is trained independently of how well the model performs on the collected.
>
> A practical instantiation of this idea is the consistency loss where we ensure consistency between the predictions from the dynamics model and the actual observations from the environment and this simple baseline works surprisingly well compared to the state of the art methods (as demonstrated by our experiments) and that others have not tried it before. Applying consistency constraint means we have two learning signals for the policy: The one from the reinforcement learning loss (i.e maximize return) and the other due to consistency constraint. We show that adding the proposed consistency constraint helps the agent to learn better dynamics model and as well as better policy for both observation space models and state space models. We compare against strong baselines:
>
> Hybrid model-based and model-free (Mb-Mf) algorithm (Nagabandi et al [1])
> Learning and Querying Fast Generative Models for Reinforcement Learning (Buesing et al [2]) - This is a state-of-the-art model for state space models.
>
> Our evaluation protocol considers a total of 7 environments and we show that using the consistency constraint leads to better generative models (in terms of log likelihood) and more powerful policy (average return) for all the cases. All the experiments are averaged over 3 random seeds and are plotted with 1 standard deviation interval.
>
> Our key contribution is the proposal of using the consistency loss which helps to learn more powerful policy and better dynamics model (as demonstrated over different tasks) while being very easy to integrate with existing model-based RL approaches. While our method is relatively simple, we are not aware of prior works that show something similar, and we believe such a simple baseline would be useful for anyone who’s working on model-based RL. Further, our experiment demonstrates the effectiveness of the approach. If we are mistaken regarding prior works, please let us know!
>
> We would like to emphasize that our work presents an extensive comparative evaluation, and we believe that these results should be taken into consideration in evaluating our work. We compare multiple approaches across more than 5 simulated tasks to the state of the art methods. Hopefully, our clarifications are convincing in terms of explaining why the evaluation is fair and rigorous, and we would, of course, be happy to modify it as needed. But at a higher level, the fact that such simple model-based approaches work better than somewhat complex model-free approaches actually is the point of the paper to me.
>
> Continued...

---

> ### Author Response · Authors · 2018-11-15
> **Response to Reviewer 3 - Part 2**
>
> How is it different from just learning a model based on k-step prediction?
> ========
>
> Our approach is different from just learning a k-step prediction model as in our case, the agent’s behavior (i.e the agent's policy) is dependent on its internal model too. In the standard case, the policy is optimized only using the RL gradient i.e maximizing expected reward and the state transition pairs (collected as the agent acts in the environment) become the supervising dataset for learning the model, and hence the policy is not affected when the model is being updated and there is no feedback from the model learning process to the policy. Hence, the data used for training the model is coming from a policy which is trained independently of how well the model performs on the collected trajectories. So the process of learning the model has no control over what kind of data is produced for its training.
>
> We propose to train both the policy and the model during the open loop. Hence the k-step predictions are used for training both the model and the policy simultaneously. Training the policy on both the RL loss and the consistency loss provides a mechanism where learning a model, can itself change the policy thus leading to a much closer interplay between the dynamics model and the policy. We show that this relatively simple approach leads to much better performance when compared to very strong baselines for both observation space and state space models for all the 7 environments we considered.
>
>
>
> ===================================================================================================
>
>
> We now refer to the specific aspects of the reviews:
>
>
> "but never follows through on the motivation of experimentation---taking actions mainly for the purpose of learning an improved dynamics model. AND Failed to connect presented work with the motivation."
>
> Our goal and motivation is to provide a mechanism for the agent to learn a better dynamics model as well as more powerful policy by ensuring the consistency in their predictions (such that predictions from the model are grounded in the real environment).
>
> This mechanism enables the agent to have a direct “interaction” b/w the agent’s policy and its dynamics model. This interaction is different from the standard approaches in reinforcement learning where the agent uses the policy to sample trajectories over which the agent is trained, and then use these sampled trajectories to learn the dynamics model. In those cases, there is no (direct) mechanism for the dynamics model to affect the policy, and hence there is no “direct interaction” between the policy and the dynamics model.  In our case, both the policy and the model are trained jointly while making sure that the predictions from the dynamics model are consistent with the observation from the environment. This provides a mechanism where learning a model can itself change the policy (thus “interacting” with the environment) instead of just training on the data coming from a policy which is trained independently of how well the model performs on the collected.
>
> A practical instantiation of this idea is the consistency loss where we ensure consistency between the predictions from the dynamics model and the actual observations from the environment and this simple baseline works surprisingly well compared to the state of the art methods (as demonstrated by our experiments) and that others have not tried it before. Applying consistency constraint means we have two learning signals for the policy: The one from the reinforcement learning loss (i.e maximize return) and the other due to consistency constraint. We show that adding the proposed consistency constraint helps the agent to learn better dynamics model and as well as better policy for both observation space models and state space models.
>
> Our evaluation protocol consists of 7 environments (Ant, Half Cheetah, Humanoid etc) and both observation space and state space models. For the observation space model, we use the “Hybrid model-based and model-free (Mb-Mf) algorithm” (Nagabandi et al [1])  which is a very strong baseline and for the state space models, we use the “Learning and Querying Fast Generative Models for Reinforcement Learning” (Buesing et al [2])  as the baseline. This model is a state-of-the-art model for state space models. We show that adding the consistency constraint to the baseline models results in improvements to both the dynamics models and the policy for all the environments that we consider.
>
> Continued...

---

> ### Author Response · Authors · 2018-11-15
> **Response to Reviewer 3 - Part 3**
>
>
> "All of their experiments merely take actions that are best according to the usual model-based or model-free methods and show that their consistency constraint allows them to learn a better dynamics model, which is not at all surprising."
>
> Our key contribution is the proposal of using the consistency loss which helps to learn more powerful policy AND better dynamics model (as demonstrated over different tasks) while being very easy to integrate with existing model-based RL approaches.  It is important to note that our proposed approach improves over the state of the art results despite being relatively simple. We are not aware of work in RL which describes and validates the benefits of imposing the consistency constraint and would be happy to include references to such work.
>
> We would like to highlight that our evaluation shows that the agent learns both better dynamics models AND more powerful policy (figure 2, 3, 5).  There seems to be some confusion about our evaluation protocol. We have updated the paper to improve that. Section 3.1 describes the different loss components and how the consistency constraint can be applied in the general. Section 3.2 and 3.3 describes the baselines and how these baselines were modified to support the consistency constraint for the observation space and the state space models respectively.
>
>
> We summarize the baselines and the evaluation protocol here:
>
> Our evaluation protocol consists of 7 environments (Ant, Half Cheetah, Humanoid etc) and both observation space and state space models. Solving Half Cheetah environment, when observations are in the pixel space (images), is very challenging as useful information like velocity is not available.
>
> For the observation space model, we use the “Hybrid model-based and model-free (Mb-Mf) algorithm” (Nagabandi et al [1]). It is a strong baseline where the authors proposed to use a trained, deep neural network based dynamics model to initialize a model-free learning agent to combine the sample efficiency of model-based approaches with the high task-specific performance of model-free methods. For the state space models, we use the “Learning and Querying Fast Generative Models for Reinforcement Learning” (Buesing et al [2])  as the baseline. This is a state-of-the-art model for state space models. As shown by our experiments (section 5), by having this consistency constraint we outperform both these baselines.
>
> We focus on evaluating the agent for both dynamics models (in terms of imagination log likelihood) and policy (in terms of average episodic returns and loss, figure 2, 3, 5). We show that adding the consistency constraint to the baseline models results in improvements to both the dynamics models and the policy for all the environments that we consider. All the experiments are averaged over 3 random seeds and are plotted with 1 standard deviation interval.
>
> Our key contribution is the proposal of using the consistency loss which helps to learn more powerful policy and better dynamics model (as demonstrated over different tasks) while being very easy to integrate with existing model-based RL approaches. While the proposed approach looks relatively simple, we are not aware of work in RL which describes and validates the benefits of imposing the consistency constraint.
>
> We would appreciate it if the reviewer could take another look at our changes and additional results, and let us know if the reviewer has request for additional changes that would alleviate the reviewer's concerns.
>
> [1]: Neural Network Dynamics for Model-Based Deep Reinforcement Learning with Model-Free Fine-Tuning - https://arxiv.org/pdf/1708.02596.pdf
>
> [2]: Learning and Querying Fast Generative Models for Reinforcement Learning - https://arxiv.org/pdf/1802.03006.pdf

---

> ### Author Response · Authors · 2018-11-19
> **Motivation**
>
> We have updated the paper, and the basic motivation is that now (as the reviewer 2 points out) the policy is updated using two learning signals - the RL loss, and the loss from the consistency constraint.  The dynamics model is not used for action selection, but only as an additional learning signal for the policy. Learning a good dynamics model is a nice side product, but this model is not used at test time.
>
> We would appreciate it if the reviewer could take another look at our changes and additional results, and let us know if the reviewer would like to request additional changes that would alleviate reviewers concerns. We hope that our updates to the manuscript address the reviewer's concerns about clarity, and we hope that the discussion above addresses the reviewer's concerns about empirical significance. We once again thank the reviewer for the feedback of our work.

---

> ### Author Response · Authors · 2018-11-20
> **Added results to evaluate the robustness of the model**
>
> Dear Reviewer
>
> We have added new evaluation results to investigate the robustness of the proposed approach in terms of compounding errors. When we use the recurrent dynamics model for prediction, the ground-truth sequence is not available for conditioning. This leads to problems during sampling as even small prediction errors can compound when sampling for a large number of steps.  We evaluate the proposed model for robustness by predicting the future for much longer timesteps (50 timesteps) than it was trained on (10 timesteps). More generally, in figure 9 (section 7.3 in appendix), we demonstrate that this auxiliary cost helps to learn a better model with improved long-term dependencies by using a training objective that is not solely focused on predicting the next observation, one step at a time.
>
> Thank you for your time! The authors appreciate the time reviewers have taken for providing feedback. which resulted in improving the presentation of our paper. Hence,  we would appreciate it if the reviewers could take a look at our changes and additional results, and let us know if they would like to either revise their rating of the paper or request additional changes that would alleviate their concerns.

---

> ### Author Response · Authors · 2018-11-30
> **More feedback ?**
>
> Thank you again for the thoughtful review. We would like to know if our rebuttal adequately addressed your concerns. We would also appreciate any additional feedback on the revised paper. Are there any other aspects of the paper that you think could be improved?

---

### Official Review · AnonReviewer1 · 2018-11-07
**A small idea, with poor comparisons**

**Rating:** 3
**Confidence:** 5

**Review:**


Summary:

This paper presents a simple auxiliary loss term for model-based RL that attempts to enforce consistency between observed experience trajectories and hallucinated rollouts.  Simple experiments demonstrate that the constraint slightly improves performance.

Quality:

While I think the idea of a consistency constraint is probably reasonable, I consider this a poorly executed exploration of the idea.  The paper makes no serious effort to compare and contrast this idea with other efforts at model-based RL.  The most glaring omission is comparison to very old ideas (such as dyna) and new ideas (such as imagination agents), both of which they cite.

Clarity:

The paper is reasonably clear, although there are some holes.  For example, in the experimental section, it is unclear what model-based RL algorithm is being used, and how it was modified to support the consistency constraint.  (I did not read the appendix).

Originality:

It is not clear how novel the central idea is.

Significance:

This idea is not significant.

Pros:
+ A simple, straightforward idea
+ A good topic - progress in model-based RL is always welcome

Cons:
- Unclear how this is significantly different from other related work (such as imagination agents)
- Experimental setup is poorly executed.
  - Statistical significance of improvements is unclear
  - No attempt to relate to any other method in the field
  - No explanation of what algorithms are being used

---

> ### Author Response · Authors · 2018-11-15
> **Response to Reviewer 1 - Part 1**
>
> We thank the reviewer for the feedback. We have conducted additional experiments to address the concerns raised about the evaluation, and we clarify specific points below. We believe that these additions address all of your concerns about the work, though we would appreciate any additional comments or feedback that you might have. We acknowledge that the paper was certainly lacking polish and accept that this may have made the paper difficult to read in places. We have uploaded a revised version in which we have revised the problem statement and writing as per the reviewer's suggestions. We briefly summarize the key idea of the paper and then address the specific concerns.
>
> What is the idea?
> =====================================
>
> Our goal is to provide a mechanism for the agent to lean a better dynamics model as well as more powerful policy by ensuring the consistency in their predictions (such that predictions from the model are grounded in the real environment).
>
> This mechanism enables the agent to have a direct “interaction” b/w the agent’s policy and its dynamics model. This interaction is different from the standard approaches in reinforcement learning where the agent uses the policy to sample trajectories over which the agent is trained, and then use these sampled trajectories to learn the dynamics model. In those cases, there is no (direct) mechanism for the dynamics model to affect the policy, and hence there is no “direct interaction” between the policy and the dynamics model.  In our case, both the policy and the model are trained jointly while making sure that the predictions from the dynamics model are consistent with the observation from the environment. This provides a mechanism where learning a model can itself change the policy (thus “interacting” with the environment) instead of just training on the data coming from a policy which is trained independently of how well the model performs on the collected.
>
> A practical instantiation of this idea is the consistency loss where we ensure consistency between the predictions from the dynamics model and the actual observations from the environment and this simple baseline works surprisingly well compared to the state of the art methods (as demonstrated by our experiments) and that others have not tried it before. Applying consistency constraint means we have two learning signals for the policy: The one from the reinforcement learning loss (i.e maximize return) and the other due to consistency constraint. We show that adding the proposed consistency constraint helps the agent to learn better dynamics model and as well as better policy for both observation space models and state space models. We compare against strong baselines:
>
> Hybrid model-based and model-free (Mb-Mf) algorithm (Nagabandi et al [1])
> Learning and Querying Fast Generative Models for Reinforcement Learning (Buesing et al [2]) - This is a state-of-the-art model for state space models.
>
> Our evaluation protocol considers a total of 7 environments and we show that using the consistency constraint leads to better generative models (in terms of log likelihood) and more powerful policy (average return) for all the cases. All the experiments are averaged over 3 random seeds and are plotted with 1 standard deviation interval.
>
> Our key contribution is the proposal of using the consistency loss which helps to learn more powerful policy and better dynamics model (as demonstrated over different tasks) while being very easy to integrate with existing model-based RL approaches. While our method is relatively simple, we are not aware of prior works that show something similar, and we believe such a simple baseline would be useful for anyone who’s working on model-based RL. Further, our experiment demonstrates the effectiveness of the approach. If we are mistaken regarding prior works, please let us know!
>
> We would like to emphasize that our work presents an extensive comparative evaluation, and we believe that these results should be taken into consideration in evaluating our work. We compare multiple approaches across more than 5 simulated tasks to the state of the art methods. Hopefully, our clarifications are convincing in terms of explaining why the evaluation is fair and rigorous, and we would, of course, be happy to modify it as needed. But at a higher level, the fact that such simple model-based approaches work better than somewhat complex model-free approaches actually is the point of the paper to me.
>
> Continued...

---

> ### Author Response · Authors · 2018-11-15
> **Response to Reviewer 1 - Part 2**
>
>
> What are the empirical results?
> =====================================
>
> Our evaluation protocol consists of 7 environments (Ant, Half Cheetah, Humanoid etc) and both observation space and state space models. Solving Half Cheetah environment, when observations are in the pixel space (images), is very challenging as useful information like velocity is not available.
>
> For the observation space model, we use the “Hybrid model-based and model-free (Mb-Mf) algorithm” (Nagabandi et al [1]). It is a strong baseline where the authors proposed to use a trained, deep neural network based dynamics model to initialize a model-free learning agent to combine the sample efficiency of model-based approaches with the high task-specific performance of model-free methods. For the state space models, we use the “Learning and Querying Fast Generative Models for Reinforcement Learning” (Buesing et al [2])  as the baseline. This is a state-of-the-art model for state space models. As shown by our experiments (section 5), by having this consistency constraint we outperform both these baselines.
>
> We focus on evaluating the agent for both dynamics models (in terms of imagination log likelihood, figure 4) and policy (in terms of average episodic returns and loss, figure 2, 3, 5). We show that adding the consistency constraint to the baseline models results in improvements to both the dynamics models and the policy for all the environments that we consider. All the experiments are averaged over 3 random seeds and are plotted with 1 standard deviation interval.
>
> ==================================================================================================
>
> We now refer to the specific aspects of the reviews:
>
> "This paper presents a simple auxiliary loss term for model-based RL that attempts to enforce consistency between observed experience trajectories and hallucinated rollouts.  Simple experiments demonstrate that the constraint slightly improves performance."
>
> Thanks for the very useful feedback. We have conducted additional experiments to address the concerns raised about the evaluation, and we clarify specific points below. We believe that these additions address all of your concerns about the work, though we would appreciate any additional comments or feedback that you might have.
>
>
> =====================================
>
> "Why is different from just learning a model based on k-step prediction?"
> "Unclear how this is significantly different from other related work (such as imagination agents)"
>
> Our approach is different from just learning a k-step prediction model as in our case, the agent’s behavior (i.e the agent's policy) is dependent on its internal model too. In the standard case, the policy is optimized only using the RL gradient i.e maximizing expected reward and the state transition pairs (collected as the agent acts in the environment) become the supervising dataset for learning the model, and hence the policy is not affected when the model is being updated and there is no feedback from the model learning process to the policy. Hence, the data used for training the model is coming from a policy which is trained independently of how well the model performs on the collected trajectories. So the process of learning the model has no control over what kind of data is produced for its training.
>
> We propose to train both the policy and the model during the open loop. Hence the k-step predictions are used for training both the model and the policy simultaneously. Training the policy on both the RL loss and the consistency loss provides a mechanism where learning a model, can itself change the policy thus leading to a much closer interplay between the dynamics model and the policy. We show that this relatively simple approach leads to much better performance when compared to very strong baselines for both observation space and state space models for all the 7 environments we considered.
>
> We have updated the paper (section 3) to describe the baselines and how to modifiy the baselines for applying the consistency constraint for both the observation space models (section 3.2) and the state space models (section 3.3).
> Experiments (Section 5) shows the improvement that result by the use of consistency constaint for both observation space models (figure 2, 3) and state space models (figure 4, 5)
>
> Continued

---

> ### Author Response · Authors · 2018-11-15
> **Response to Reviewer 1 - Part 3**
>
>
>
> "no serious effort to compare and contrast this idea with other efforts at model-based RL.  … it is unclear what model-based RL algorithm is being used, and how it was modified to support the consistency constraint."
>
> We have updated the paper to address the concern about the baselines and the proposed approach not being described in detail. Section 3.1 describes the different loss components and how the consistency constraint can be applied in the general. Section 3.2 and 3.3 describes the baselines and how these baselines were modified to support the consistency constraint for the observation space and the state space models respectively.
>
> We summarize the baselines and the evaluation protocol here:
>
> Our evaluation protocol consists of 7 environments (Ant, Half Cheetah, Humanoid etc) and both observation space and state space models. For the observation space model, we use the “Hybrid model-based and model-free (Mb-Mf) algorithm” (Nagabandi et al [1])  which is a very strong baseline and for the state space models, we use the “Learning and Querying Fast Generative Models for Reinforcement Learning” (Buesing et al [2])  as the baseline. This model is a state-of-the-art model for state space models.
>
> We focus on evaluating the agent for both dynamics models (in terms of imagination log likelihood, figure 4) and policy (in terms of average episodic returns and loss, figure 2, 3, 5). We show that adding the consistency constraint to the baseline models results in improvements to both the dynamics models and the policy for all the environments that we consider. All the experiments are averaged over 3 random seeds and are plotted with 1 standard deviation interval.
>
> =====================================
>
>
> "It is not clear how novel the central idea is."
>
> Our key contribution is the proposal of using the consistency loss which helps to learn more powerful policy and better dynamics model (as demonstrated over different tasks) while being very easy to integrate with existing model-based RL approaches. While our method is relatively simple, we are not aware of prior works that show something similar, and we believe such a simple baseline would be useful for anyone who’s working on model-based RL. Further, our experiment demonstrates the effectiveness of the approach. If we are mistaken regarding prior works, please let us know!
>
> We would like to emphasize that our work presents an extensive comparative evaluation, and we believe that these results should be taken into consideration in evaluating our work. We compare multiple approaches across more than 5 simulated tasks to the state of the art methods. Hopefully, our clarifications are convincing in terms of explaining why the evaluation is fair and rigorous, and we would, of course, be happy to modify it as needed. But at a higher level, the fact that such simple model-based approaches work better than somewhat complex model-free approaches actually is the point of the paper to me.
>
>
> =====================================
>
>
> "Statistical significance of improvements is unclear"
>
> Our evaluation protocol (section 5) consists of 7 environments (Ant, Half Cheetah, Humanoid etc) and both observation space and state space models. Solving Half Cheetah environment, when observations are in the pixel space (images), is very challenging as useful information like velocity is not available.
>
>
> For the observation space model (section 5.1), we use the “Hybrid model-based and model-free (Mb-Mf) algorithm” (Nagabandi et al [1]). It is a very strong baseline where the authors proposed to use a trained, deep neural network based dynamics model to initialize a model-free learning agent to combine the sample efficiency of model-based approaches with the high task-specific performance of model-free methods. For the state space models (section 5.2), we use the “Learning and Querying Fast Generative Models for Reinforcement Learning” (Buesing et al [2])  as the baseline. This is a state-of-the-art model for state space models.
>
> We focus on evaluating the agent for both dynamics models (in terms of imagination log likelihood) (figure 4) and policy (in terms of average episodic returns) (figure 2, 3, 5). We show that adding the consistency constraint to the baseline models results in improvements to both the dynamics models and the policy for all the environments that we consider. All the experiments are averaged over 3 random seeds and are plotted with 1 standard deviation interval.
>
> We would appreciate it if the reviewer could take another look at our changes and additional results, and let us know if the reviewer has a request for additional changes that would alleviate the reviewer's concerns.
>
> [1]: Neural Network Dynamics for Model-Based Deep Reinforcement Learning with Model-Free Fine-Tuning - https://arxiv.org/pdf/1708.02596.pdf
>
> [2]: Learning and Querying Fast Generative Models for Reinforcement Learning - https://arxiv.org/pdf/1802.03006.pdf

---

> ### Author Response · Authors · 2018-11-20
> **Added results to evaluate the robustness of the model**
>
> Dear Reviewer
>
> We have added new evaluation results to investigate the robustness of the proposed approach in terms of compounding errors. When we use the recurrent dynamics model for prediction, the ground-truth sequence is not available for conditioning. This leads to problems during sampling as even small prediction errors can compound when sampling for a large number of steps.  We evaluate the proposed model for robustness by predicting the future for much longer timesteps (50 timesteps) than it was trained on (10 timesteps). More generally, in figure 9 (section 7.3 in appendix), we demonstrate that this auxiliary cost helps to learn a better model with improved long-term dependencies by using a training objective that is not solely focused on predicting the next observation, one step at a time.
>
> Thank you for your time! The authors appreciate the time reviewers have taken for providing feedback. which resulted in improving the presentation of our paper. Hence,  we would appreciate it if the reviewers could take a look at our changes and additional results, and let us know if they would like to either revise their rating of the paper or request additional changes that would alleviate their concerns.

---

> ### Author Response · Authors · 2018-11-30
> **More Feedback ?**
>
> Thank you again for the thoughtful review. We would like to know if our rebuttal adequately addressed your concerns. We would also appreciate any additional feedback on the revised paper. Are there any other aspects of the paper that you think could be improved?

---

### Author Response · Authors · 2018-11-30
**Rebuttal Summary**

We briefly summarize the reviewers' comments and describe how we address that:

* Reviewer 1 pointed we are missing some comparisons (which we have highlighted while editing the paper). The clarity issues have been addressed as well. Regarding the idea being "significant", we highlight that at a higher level, the fact that a simple model-based approaches work better than somewhat complex model-free approaches actually is the point of the paper. We compare multiple approaches across more than 5 simulated tasks to the state of the art methods and our experiment demonstrates the effectiveness of the approach and we believe such a simple baseline would be useful for anyone who's working on model-based RL.

* Reviewer 2 provided a very thorough review which we incorporated in our updated version and the reviewer increased our scores to 5. The reviewer had some reservations based on the significance of the results. To that end, we conducted extra experiments to evaluate the robustness of the k-step unrolled model as well.

* Reviewer 3 highlighted that baselines methods need to be elaborated and to that end, we highlight that for our method, the policy is updated using two learning signals - the RL loss, and the loss from the consistency constraint. This is the key for why the method works. The dynamics model is not used for action selection, but only as an additional learning signal for the policy, and hence learning a good dynamics model is a nice side product, but this model is not used at test time. The other issues related to writing have been addressed.

---

### Author Response · Authors · 2018-11-30
**A simple baseline works very well**

We thank the reviewers and the ACs for taking the time to go through our work. The initial reviews highlighted the need to improve the clarity of the paper (a better description of the proposed model,  experiments etc). It also led to some confusion about how useful and relevant our baselines were. We acknowledge that the paper was certainly lacking polish and accept that this may have made the paper difficult to read in places. We updated the paper, improved the description of the model and the experiments and addressed the concerns raised, We also performed additional experiments to highlight the robustness of our model for multi-step prediction.  We  briefly summarize the key idea of the paper and note how is it different from existing works

What is the idea?
========
Using the consistency loss which helps to learn more powerful policy AND better dynamics model (as demonstrated over different *7* tasks) while being very easy to integrate with existing model-based RL approaches.

Isnt it too simple?
===========

At a higher level, the fact that a simple model-based approaches work better than somewhat complex model-free approaches actually is the point of the paper. We compare multiple approaches across more than 5 simulated tasks to the state of the art methods and our experiment demonstrates the effectiveness of the approach and we believe such a simple baseline would be useful for anyone who's working on model-based RL.

Why does it work?
========
Training the policy on both the RL loss and the consistency loss provides a mechanism where learning a model, can itself change the policy thus leading to a much closer interplay between the dynamics model and the policy.


How is it different from just learning a model based on k-step prediction?
========
Iin our case, the agent's behavior (i.e the agent's policy) is dependent on its internal model too. In the baseline case, the state transition pairs (collected as the agent acts in the environment) become the supervising dataset for learning the model, and the process of learning the model has no control over what kind of data is produced for its training.


How to implement it?
========
Impose a consistency loss to ensure consistency between the predictions from the dynamics model and the observations from the enviornment. Train both the policy and the model simultaneously during the open loop.


How good are the results?
========

Our evaluation protocol consists of 7 environments (Ant, Half Cheetah, Humanoid etc) and both observation space and state space models. Solving Half Cheetah environment, when observations are in the pixel space (images), is very challenging as useful information like velocity is not available.

For the observation space model, we compare against the "Hybrid model-based and model-free (Mb-Mf) algorithm" (Nagabandi et al). and for the state space models, we compare against "Learning and Querying Fast Generative Models for Reinforcement Learning" (Buesing et al [2]) (SOTA for state space models). As shown by our experiments (section 5), by having this consistency constraint we outperform both these baselines.

We focus on evaluating the agent for both dynamics models (in terms of imagination log likelihood, figure 4) and policy (in terms of average episodic returns and loss, figure 2, 3, 5). We show that adding the consistency constraint to the baseline models results in improvements to both the dynamics models and the policy for all the environments that we consider. All the experiments are averaged over 3 random seeds and are plotted with 1 standard deviation interval.

---

### Meta-Review · Area_Chair1 · 2018-12-14

**Confidence:** 4
**Recommendation:** Reject

**Metareview:**

The paper proposes and approach for model-based reinforcement learning that adds a constraint to encourage the predictions from the model to be consistent with the observations from the environment. The reviewers had substantial concerns about the clarify of the initial submission, which has been significantly improved in revisions of the paper. The experiments have also been improved.
Strengths: The method is simple, the performance is competitive with state-of-the-art approaches, and the experiments are thorough including comparisons on seven different environments.
Weaknesses: The main concern of the reviewers is the lack of concrete discussion about how the method compares to prior work. While the paper cites many different prior methods, the paper would be significantly improved by explicitly comparing and contrasting the ideas presented in this paper and those presented in prior work. A secondary weakness is that, while the results appear to be statistically significant, the improvement over prior methods is still relatively small.
I do not think that this paper meets the bar for publication without an improved discussion of how this work is placed among the existing literature and without more convincing results.

As a side note, the authors should consider comparing to the below NeurIPS '18 paper, which significantly exceeds the performance of Nagabandi et al '17: https://arxiv.org/abs/1805.12114